# Semantic-aware Next-Best-View for Multi-DoFs Mobile System in Search-and-Acquisition based Visual Perception

## ABSTRACT

Efficient visual perception using mobile systems is crucial, particularly in unknown environments such as search and rescue operations, where swift and comprehensive perception of objects of interest is essential. In such real-world applications, objects of interest are often situated in complex environments, making the selection of the 'Next Best' view based solely on maximizing visibility gain suboptimal. Semantics, providing a higher-level interpretation of perception, should significantly contribute to the selection of the next viewpoint for various perception tasks. In this study, we formulate a novel information gain that integrates both visibility gain and semantic gain in a unified form to select the semantic-aware Next-Best-View. Additionally, we design an adaptive strategy with termination criterion to support a two-stage search-and-acquisition manoeuvre on multiple objects of interest aided by a multi-degree-of-freedoms (Multi-DoFs) mobile system. Several semantically relevant reconstruction metrics, including perspective directivity and region of interest (ROI)-to-full reconstruction volume ratio, are introduced to evaluate the performance of the proposed approach. Simulation experiments demonstrate the advantages of the proposed approach over existing methods, achieving improvements of up to 27.13% for the ROI-to-full reconstruction volume ratio and a 0.88234 average perspective directivity. Furthermore, the planned motion trajectory exhibits better perceiving coverage toward the target.

## CCS CONCEPTS

• **Computing methodologies** → **Planning and scheduling**; Graphics systems and interfaces; • **Information systems** → *Retrieval tasks and goals*.

## KEYWORDS

Mobile platform visual acquisition, Next-Best-View, Semantics

## 1 INTRODUCTION

Efficient visual acquisition is a crucial aspect of unknown scene perception using mobile platforms, providing essential information for various manipulation tasks, such as search and rescue operations. Multi-DoFs mobile systems equipped with cameras have become increasingly popular due to their high mobility and agility,

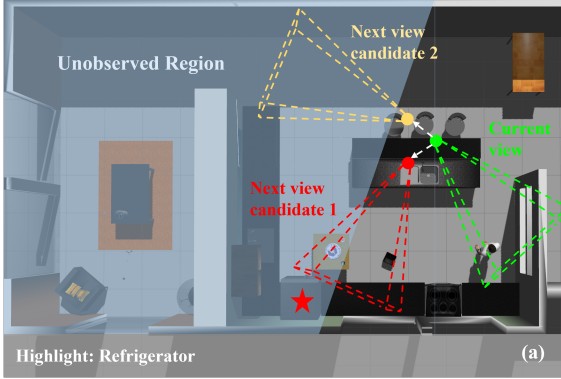

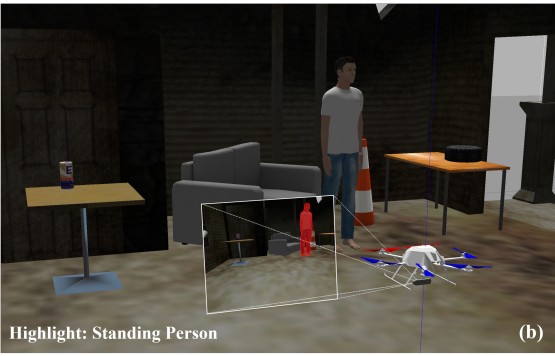

**Figure 1: (a) When the refrigerator is designated as the object of interest, next view candidate 1 provides higher semantic gain while next view candidate 2 offers higher visibility gain; (b) A capture of observing a view with high semantic gain in the experiment, with the standing person is highlighted.**

making them well-suited for a wide range of applications. Specifically, autonomous visual acquisition by multi-DoF mobile systems (e.g. unmanned aerial vehicles, UAVs) in unknown and inaccessible environments has proven to be an effective means in search and rescue, reducing the need for professional remote control skills among emergency personnel. However, exhaustive observation is a time-consuming and resource-intensive process. To ensure an efficient visual perception process, it is vital to select adaptive views that provide the most information. Next-Best-View (NBV) was initially presented for an unknown area exploration using the mobile robot [3, 25, 34, 38], usually a finite iteration random tree is grown in the known free space, e.g., Rapidly-exploring Random Tree (RRT), RRT* [17, 22] then the best branch is selected by maximizing the gain (e.g., the amount of unobserved space that can be observed) while minimizing the moving cost (e.g., distance or time cost). After that, it was also adopted to the path planning for single object surface reconstruction [20, 21], online inspection [27, 35, 36] and so on.

However, the existing studies determining the next best view focus on information gain by evaluating the visibility of unknown voxels, regardless of their semantics. Unlike the previously mentioned scenarios, visual perception on the objects of interest under complex environments should be semantically selective rather than solely focused on perceiving the unknowns. In other words, the "Next Best" viewpoint in a complex environment cannot be evaluated effectively without the relevant semantic information. In Figure 1(a), semantically informative views should be selected as a higher priority to ensure the efficiency of visual perception on the specific target using mobile systems.

In this work, we propose a semantic-aware NBV scheme for efficient visual perception under complex environments and implement it in a two-stage search-and-acquisition manoeuvre aided by the multi-DoFs mobile system. We develop a novel information gain formulation which integrates both semantic gain and visibility gain. We also design an adaptive strategy to balance these two components so that the mobile robot can perform both search and acquisition operations on specified semantically important objects. We evaluate the proposed approach using different self-build scenarios in the simulation environment; an experiment capture is shown in Figure1(b). The results we obtained demonstrate that the proposed approach significantly improves the efficiency of visual perception on specified objects under complex environments through evaluating the reconstruction progress against region of interest (ROI) in volume, ROI-to-full reconstruction volume ratio and perspective directivity. Both the motion planning and reconstruction are implemented based on the voxblox [29] as the map representation, which employs Truncated Signed Distance Fields (TSDFs) to represent the object surface. Then, the RRT* is generated in the observed free space. To the best of our knowledge, this is the first work that investigates semantic-aware NBV for search-and-acquisition-based visual perception by mobile systems, which integrates the contribution from both semantic gain and visibility gain in a unified form for evaluating and selecting the next viewpoint. We demonstrate its capability in the application of different complex environments.

The main contributions of this work include:

(1) We present a novel information gain formulation for evaluating the candidate viewpoints that integrates both semantic gain and visibility gain. Such novel formulation can be applied to many other application scenarios in which the visual data acquired contain rich semantics of the complex environment.

(2) We design an adaptive strategy with termination criterion to balance the semantic and visibility terms so that the mobile platform can perform an effective two-stage search-and-acquisition manoeuvre on the specified object or multiple objects under the complex environment. The principle behind this two-stage approach can also be applied to scenarios in which the objective of the task can be properly decomposed to facilitate effective implementation.

(3) To assess this novel formulation, we also introduce several evaluation metrics to characterize the system performance and demonstrate the efficiency in perceiving the specific objects under the complex environment while the data acquisition mobile system is undergoing multi-DoFs motion.

The paper content is organized as follows: an overview of the related work and how we step further is presented in Section 2. We introduce the proposed system and showcase its effectiveness in visual acquisition on the objects of interest in simulation experiments in Sections 3 and 4. Finally, we analyze the results obtained and draw conclusions in Sections 5 and 6.

## 2 RELATED WORK

### 2.1 Mobile System Informative Path Planning for Visual Acquisition

Real-time informative path planning is typically the approach to tackle the non-model-based visual acquisition problem that has no prior information or knowledge of the environment or the target object. Thus, the non-model-based reconstruction needs to plan each view in real-time, which is different from the model-based approach that can be planned offline. There are two main approaches for evaluating new viewpoints in 3D reconstruction: surface-based methods and volumetric methods. Surface-based approaches represent the 3D shape as a mesh and evaluate new views by analyzing the mesh surface [5]. For example, Krainin et al. [17] used a surface-based approach that modelled uncertainty with a Gaussian distribution along each camera ray and measured information gain as the total entropy reduction weighted by surface area. Surface-based methods can evaluate the quality of the 3D model during reconstruction but are computationally expensive due to complex visibility calculations [33]. And more recently, dynamic objects can be accurately reconstructed by surface-based method [37]. Volumetric methods, on the other hand, represent the 3D shape with voxels, which allow for simple visibility calculations and estimating the probability that each voxel is occupied [14]. Volumetric view evaluation casts rays from the candidate next views through the voxel space to simulate how a camera would sample the scene. Volumetric approaches are computationally more efficient but may not directly provide a surface model of the 3D shape. After that, hybrid methods [21] have combined both surface and volumetric representations to gain the benefits of each. In summary, surface-based 3D reconstruction evaluates new views by analyzing an estimated 3D mesh surface [5, 19], while volumetric methods evaluate new views by casting rays through the voxel representation [14]. The hybrid methods use both representations to improve the efficiency of 3D modelling [21].

### 2.2 Next-Best-View and Related Applications

Next-Best-View is a widely-used greedy method to find local solutions from incomplete information. It was first addressed in the 1980s [7, 24]. It determines the next viewpoint that can observe the largest information gain from the current map iteratively, finally resulting in a completed observation. The information gain metric depends on the specific application and requirements. In order to perceive the unknown volumetric information, besides the early stage approach [1] which simply counts the number of unknown voxels that can be seen, Kriegel et al. [21] use information theoretic entropy to estimate the expected observation. To achieve high completeness of reconstruction, Delmerico et al. [9] proposed proximity count and area factor volumetric information, optimizing the expected gain on a probabilistic map. In 2016, Bircher et

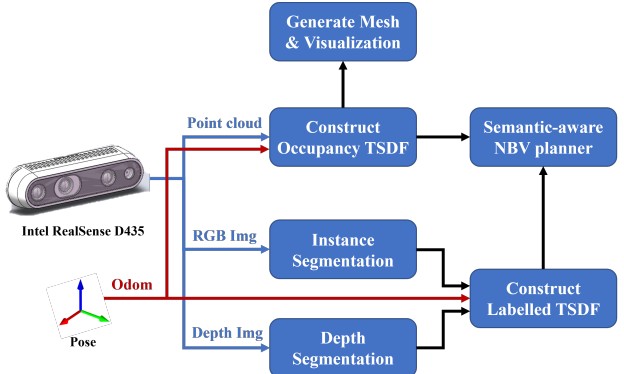

**Figure 2: Diagram of the system overview: Both the occupancy map and labelled map are constructed in parallel. The Semantic-aware NBV planner takes two maps as the input. The reconstructed mesh is visualized using the occupancy TSDF map.**

al. [3] presented the receding horizon NBV (RH-NBV) that adopts the core idea of model predictive control (MPC). It only executes the first edge in the best branch of RRT to avoid the dilemma of local minima. It also introduces the exponential discount term to penalize the long-distance path. In [32], a novel utility function is formulated as the ratio of gain and cost, minimizing the number of parameters that need to be fine-tuned. Different from the NBV, which belongs to the sampling-based informative viewpoint planning method, the frontier-based method [39] consistently pursues the boundaries between the explored free and unexplored areas in the occupancy map. The frontier-based method is widely employed in high-speed flight and fast exploration tasks [2, 6], but it is difficult to be generalized to other applications since it cannot have a flexible information gain formulation in NBV fashion. In [15], an uncertainty-guided mapless NBV scheme is proposed, leading to more accurate scene reconstruction. In [26], the predicted fruit shapes are explicitly used to compute information gain for fruit mapping and reconstruction.

Due to the simplicity of the purpose or the environment, there is no existing research that has focused on the contribution of semantics on viewpoint selection in NBV fashion in the application of either unknown exploration or single-object reconstruction. However, under challenging environments (e.g., search and rescue), searching and perceiving the semantic informative views can help us model the object and its surroundings more efficiently. For a more closely related work [18] that proposed a semantically informed scheme for reconstruction. It presents the utility term multiplied by the entropy-formed gain, but does not formulate the semantic term explicitly. It may result in penalization on the unknown exploration capability and would be difficult to generalize to different tasks such as the search–and-acquisition mission.

## 3 PROPOSED METHOD

### 3.1 Problem Description

The problem considered in this work is that there exist one or more specific targets $A = \{A_1, ..., A_N\}$, located at the unknown positions in a 3D space $V \subset \mathbb{R}^3$. Unlike the other exploration approaches, the focus is not on observing all the free and occupied space ($V_{free}$ and $V_{occ} \subset V$) to achieve $V_{free} \cup V_{occ} = V$. Instead, our approach focuses on searching for and observing each target $A_k \in A$ sequentially, We begin by exploring space $V$ and once we identify a set of occupied voxels $V_{obA-k} \subset V_{occ}$ that have been labelled as $c_{tgt-k}$, it indicates that the target $A_k$ has been found. Then the acquisition mode is initiated to retrieve not only the volume $V_{tgt-k}$ of each target $A_k$, but also its surroundings ($V_{sur-k} \subset V_{free}$ or $V_{sur-k} \subset V_{occ}$), with the objective of effectively enlarging the observed volume $V_{obA-k}$ s.t. min $|V_{res-k}| = \min |V_{tgt-k} - V_{obA-k}|$ utilizing the most extensive accessible perspective coverage within the limited time, where $V_{res-k}$ represents the residue voxels of the target $A_k$. The searching and acquisition process will be switched to the next target $A_{k+1}$ after achieving the maximum observation on $A_k$.

### 3.2 System Overview

Two maps are constructed to support the two-stage search-and-acquisition scheme of the proposed semantic-aware NBV framework, an occupancy TSDF map and a labelled TSDF map. Figure 2 illustrates the overall system, where the occupancy TSDF map is incrementally updated from the observations. This is achieved by utilizing the point cloud input from the Intel RealSense D435i depth camera and the real-time pose of the UAV, following the approach proposed in voxblox [29]. The occupancy TSDF map provides information about the occupancy status of the environment, which is essential for the planner to generate RRT* in free space and calculate visibility gain. Additionally, we constructed a labelled TSDF map inspired by the work of Grinvald et al. [12]. This map is generated by raycasting the overlap of the segmentation results from Mask R-CNN [13] based on the RGB input and the depth segments from the depth image input. Depth segments are identified by finding the convex area of depth discontinuity in the depth image. The labelled TSDF map provides a detailed representation of the environment's geometry with semantics, which is useful for the planner to identify the semantic gain. The different types of map representations used in the system are organized into separate layers, with each layer consisting of a set of blocks that are indexed based on their position in the map. It is the same as the structure adopted in voxblox [29]. The mapping between the block positions and their locations is stored in the hash table adopting voxel hashing [28]. Finally, the acquisition result is visualized by the surface model generated from the occupancy TSDF map.

### 3.3 Semantic-aware NBV Framework

From the representation of input TSDF maps, the space $V$ is divided into separate layers of unit-volume cubical voxel $m_o \in \mathcal{M}_o$, $m_l \in \mathcal{M}_l$, where $\mathcal{M}_o$ and $\mathcal{M}_l$ denote the occupancy and labelled map respectively. Each voxel $m_{oi}$ in the occupancy map $\mathcal{M}_o$ consists of an associated centre position $p_i$, distance $d_i$, weight $w_i$ and state $s_i$. The centre position is represented by $p_i$ using the coordinate of its geometric centre, and the voxel's distance from the surface boundary is represented by $d_i$. In order to minimize the quadratic sensing error of the 3D sensor (e.g., depth camera), we adopt the distance $d_i$ updating approach in [32]. The weight $w_i$ is a metric

that refers to the reliability of the distance's measurement. Here we employ the weighting method formulated in voxblox [29]. The state $s_i$ of each voxel can be marked as "FREE", "OCCUPIED", or "UNKNOWN". For the voxel $m_{li}$ in the labelled map $\mathcal{M}_l$, there are three additional associated properties instance label $l_i$, semantic category $c_i$ and label confidence $l_{ci}$. In which the instance label is the index with the highest overlap probability between the binary mask result $m_i$ from Mask R-CNN and the result $r_i$ from depth segmentation. The corresponding semantic category is assigned to $c_i$ if available; otherwise, the default semantics is the background. The label confidence is the number of times the voxel has been labelled as $l_i$ divided by the observation times.

*3.3.1 Visibility Gain Formulation.* In order to perceive the unknown area and search for the target we are interested in, we define the visibility gain of a branch $b$ associated $n$ nodes $\{b_1, b_2, ..., b_n\}$ in Equation 1.

$$Visible(\mathcal{M}_o, b) = \sum_{j}^{n} Visible(\mathcal{M}_o, b_j) \tag{1}$$

The visible voxels $\{m_{o1}, m_{o2}, ..., m_{om}\}$ at node $b_j$ are obtained using the intrinsic and extrinsic parameters of the camera. Thus,

$$Visible(\mathcal{M}_o, b_j) = \sum_{i}^{m} V\_gain(\mathcal{M}_o, m_{oi}) \tag{2}$$

For simply perceiving the unknown in the 'search' stage, we employ the conventional $V\_gain$ formulation that applies a unit increase in gain if the $s_i$ is "UNKNOWN", and there is no gain for "OCCUPIED" or "FREE" voxel.

*3.3.2 Semantic Gain Formulation.* Similar to the visibility gain, we have the semantic gain for each branch:

$$Semantic(\mathcal{M}_l, b) = \sum_{j}^{n} Semantic(\mathcal{M}_l, b_j) \tag{3}$$

Again for each node $b_j$ on the branch,

$$Semantic(\mathcal{M}_l, b_j) = \sum_{i}^{m} S\_gain(\mathcal{M}_l, m_{li}) \tag{4}$$

The $S\_gain$ for each visible voxel $m_{oi}$ at the specific node $b_j$ is formulated intuitively favours the viewpoints that can observe the new area around the labelled target voxel. As is shown in Equation 5.

$$S\_gain(\mathcal{M}_l, m_{li}) = \begin{cases} exp(-\lambda_1 c_1(d_{li})), \\ \quad if \ s_i = Unknown \\ \\ \eta_{tgt} \cdot f(m_{li}), \ if \ s_i = Occupied \\ \quad\quad\quad\quad \&\& \ c_i = c_{tgt-k} \\ \\ exp(-\lambda_2 c_2(d_{li})), \\ \quad if \ s_i = Occupied \ \&\& \ c_i! = c_{tgt-k} \\ \quad\quad \&\& \ c_i! = background \\ \\ 0, \ otherwise \end{cases} \tag{5}$$

Where $\eta_{tgt}$ denotes the influence factor that refers to the significance or priority of the voxel with the target label. The exponential term represents the exponential discount on the influence regarding the distance $d_{li}$ of the current voxel to the target volume $V_{obA-k}$ of

the target $A_k$. $\lambda_1$, $\lambda_2$ are the weight term and $c_1$, $c_2$ are the discount cost functions. In order to minimize the sensing error and refine the voxel that has already been labelled as $c_{tgt-k}$, we also introduced the function $f$ in Equation 6 as its gain.

$$f(m_{li}) = (1 - \frac{|N_{rays}(m_{li}) - N_{exp}|}{1 + |N_{rays}(m_{li}) - N_{exp}|}) \cdot (1 - \frac{w_i}{1 + w_i}) \tag{6}$$

Where $N_{rays}(m_{li})$ denotes the number of rays intersecting the $m_{li}$, which is usually proportional to the inverse of depth quadratically. $N_{exp}$ represents the expected number of intersecting rays. The list $\mathcal{L}_{tgt}$ stores the voxels which have been labelled with the semantic category $c_{tgt-k}$, and $\mathcal{L}_{tgt}$ is maintained to serve the calculation of the shortest distance $d_{li}$. Inspired by [23], we maintain the listed voxels (i.e. $V_{obA-k}$) in a continuous and convex shape.

*3.3.3 Adaptive Strategy with Termination Criterion.* The proposed method integrates both visibility gain and semantic gain in a consistent format in Equation 7.

$$Gain(\mathcal{M}_o, \mathcal{M}_l, K) = K \cdot \sum_{j}^{n} Visible(\mathcal{M}_o, b_j) f_o(\delta_{b_{j-1}}^{b_j})$$
$$+ (1-K) \cdot \sum_{j}^{n} Semantic(\mathcal{M}_l, b_j) f_l(\delta_{b_{j-1}}^{b_j}) \tag{7}$$

Where $\delta_{b_{j-1}}^{b_j}$ denotes the edge distance from node $b_{j-1}$ to node $b_j$. $K$ is a bool variable controlling the mode preference switching between 'search' and 'acquisition' in our case. $f_o$ and $f_l$ represent the cost function penalizing on the distance of the long edge. It could be in the form of exponential penalty [4, 34], linear penalty [8] or a reciprocal cost to reduce the complexity in tuning parameters [32]. Here, we employ the format in [32]. $\lambda_o$, $\lambda_l$ are the constant parameters.

$$f_o(\delta_{b_{j-1}}^{b_j}) = 1/\lambda_o \delta_{b_{j-1}}^{b_j} \tag{8}$$

$$f_l(\delta_{b_{j-1}}^{b_j}) = 1/\lambda_l \delta_{b_{j-1}}^{b_j} \tag{9}$$

The state switching of the bool variable $K$ ensures the smoothness of the stage changing between searching and acquisition in the manoeuvre. We also introduce a termination criterion for the acquisition stage to perform the target switching within a manoeuvre. We separate the semantic gain for each branch into three parts:

$$S_{unknown}(\mathcal{M}_l, b) = \sum_{b_j} \sum_{m_{li}|con1} exp(-\lambda_1 c_1(d_{li})) \tag{10}$$

$$S_{refine}(\mathcal{M}_l, b) = \sum_{b_j} \sum_{m_{li}|con2} \eta_{tgt} \cdot f(m_{li}) \tag{11}$$

$$S_{surround}(\mathcal{M}_l, b) = \sum_{b_j} \sum_{m_{li}|con3} exp(-\lambda_2 c_2(d_{li})) \tag{12}$$

Where con1 refers to condition 1 $s_i = Unknown$, con2 refers to $s_i = Occupied \ \&\& \ c_i = c_{tgt-k}$ and con3 refers to $s_i = Occupied \ \&\& \ c_i! = c_{tgt-k} \ \&\& \ c_i! = background$. The planner starts with a zero-size $\mathcal{L}_{tgt}$, $K$ is initially assigned to 1. Once the list $\mathcal{L}_{tgt}$ is expanded, $K$ is switched to 0. The acquisition for one target is terminated if $S_{surround}$ is far greater than the summation of $S_{unknown}$ and $S_{refine}$ for $c_{thre}$ branches. Then $K$ flips to 1, the $\mathcal{L}_{tgt}$ is cleared, and meanwhile, the target label is switched to $c_{tgt-(k+1)}$. Once the

list $\mathcal{L}_{tgt}$ is further expanded, $K$ will be set to 0 again. This described strategy can also be represented as Algorithm 1 below.

---

**Algorithm 1** Semantic-aware NBV adaptive strategy with termination criterion

---

  $K = 1$;
  **while** Occupancy TSDF and Labelled TSDF is updated **do**
    last_size = $\mathcal{L}_{tgt}$.size();
    Maintain the list $\mathcal{L}_{tgt}$;
    **if** $\mathcal{L}_{tgt}$.size() > 0 **then**
      Calculate $S_{unknown}, S_{target}, S_{refine}$
      **if** $\mathcal{L}_{tgt}$.size() > last_size **then**
        $K = 0$;
      **else if** $Count(S_{surround} \gg S_{unknown} + S_{refine}) > c_{thre}$
      **then**
        $K = 1$;
        $\mathcal{L}_{tgt}$.clear();
        Target switch to $c_{tgt-(k+1)}$;
      **end if**
    **end if**
    gain = $Gain(\mathcal{M}_o, \mathcal{M}_l, K)$;
  **end while**

---

# 4 EXPERIMENTS AND RESULTS

## 4.1 Experimental Setup

Since the planner operates within a perception-planning-execution loop, realistic simulation is compulsory for the evaluation of the proposed scheme. The proposed approach is tested in the simulated world scenes in Gazebo, a 3D dynamic physical robotics simulator. The developed behaviour of the UAV is operating on the Robot Operating System (ROS) [31]. Gazebo-based simulator RotorS [11] is employed to provide an accurate model of the UAV's physics. The underlying control hierarchy of UAV is presented in [16].

The experiments are conducted in the simulation environment, three different settings with two self-build scenes (a narrow collapsed scene with an uneven lighting condition and a larger indoor house scene with ideal lighting condition) in Gazebo with the aid of the individual models by Open Robotics [30] and Google Research [10]. And all the experiment results are collected on the machine with an Intel 8C16T Core i7-11700KF at 3.6 GHz × 16 and an NVIDIA GeForce RTX 3060 graphic card.

*4.1.1 Collapsed Room Scene.* The Collapsed Room Scene used in the experiment is a 10 m × 10 m × 2.5 m map with various furniture, industrial tools and a standing person within the obstacles. The standing person is highlighted as the specific target.

*4.1.2 Kitchen and Dining Room Scene.* The Kitchen and Dining Room Scene used in the experiment is a 16 m × 10 m × 3.5 m map with common facilities in the family house and a standing person in the corner. The standing person is highlighted as the specific target.

*4.1.3 Kitchen and Dining Room with Multiple Specified Objects.* The third one has the same environment as the Kitchen and Dining

**Table 1: System parameters for all experiments**

| Max. velocity | 0.8 m/s | Camera RGB FOV | $68° \times 42°$ |
|---|---|---|---|
| Max. acceleration | 0.8 m/s$^2$ | Camera depth FOV | $87° \times 58°$ |
| Max. yaw rate | $\pi/4$ rad/s | Camera ray length | 5 m |

Room Scene, but the refrigerator and sink are highlighted as the specific targets.

The basic system parameters are consistent throughout all the experiments described in this study, including the motion dynamics constraints of the UAV and the camera parameters for acquisition, as shown in Table 1. For each experiment, the UAV starts at the initial pose where the target is not within the field of view (FOV).

## 4.2 Evaluation Metrics

Since the proposed scheme is designed to execute the search-and-acquisition manoeuvre for the specific target, once the target is found, we aim to acquire the target from multiple accessible viewpoints and achieve maximum reconstruction coverage of the target itself and potential interactions with the surroundings. For most real-world applications, the perception, planning and motion control pipeline of the UAV is expected to execute fully onboard. Thus, time usage is also a crucial aspect of the evaluations.

*4.2.1 Perspective Directivity.* In order to measure whether the UAV consistently perceives the target and its nearest surroundings during the acquisition stage, we calculated the perspective directivity in the target direction $D_{tgt-k}$ for each selected view. The current position $p^i$, and orientation $o^i$ of the UAV at view $i$ are obtained from the odometry. The ground truth position of the target $p_{tgt-k}$ is a privileged knowledge that we defined based on the built scene and is not known to the planner. $o^i_P$ and $o^i_Y$ denote the pitch and yaw angle of view $i$. The directivity at the target direction $D^i_{tgt-k}$ of view $i$ is defined as the cosine of the angle between the camera optical axis $O_{cam}$ and the line connecting the current position $p^i$ with the target position $p_{tgt-k}$, i.e.

$$O^i_{cam} = [cos(o^i_P) \cdot cos(o^i_Y), cos(o^i_P) \cdot sin(o^i_Y), sin(o^i_P)] \quad (13)$$

$$D^i_{tgt} = cos < O^i_{cam}, (p_{tgt-k} - p^i) > \quad (14)$$

*4.2.2 ROI Reconstruction Progress in Volume and ROI-to-full Reconstruction Ratio.* In addition to the perspective directivity, we also periodically record the reconstructed map and analyze the global growth of the reconstruction volume as well as the growth within the region of interest. Again, the region of interest (ROI) is a privileged knowledge that is not known to the planner. It comprises the target $V_{tgt-k}$ and the nearest surroundings $V_{sur-k}$. The volume ratio of reconstructed ROI over the full reconstructed map indicates the strength of the purpose. A higher ratio implies less reconstruction redundancy in perceiving the target under the complex environment, while a lower ratio indicates more storage consumption on the non-important reconstructions. The target perceiving coverage is analyzed and compared using the motion trajectories with each pose of the UAV and the frustum of the camera.

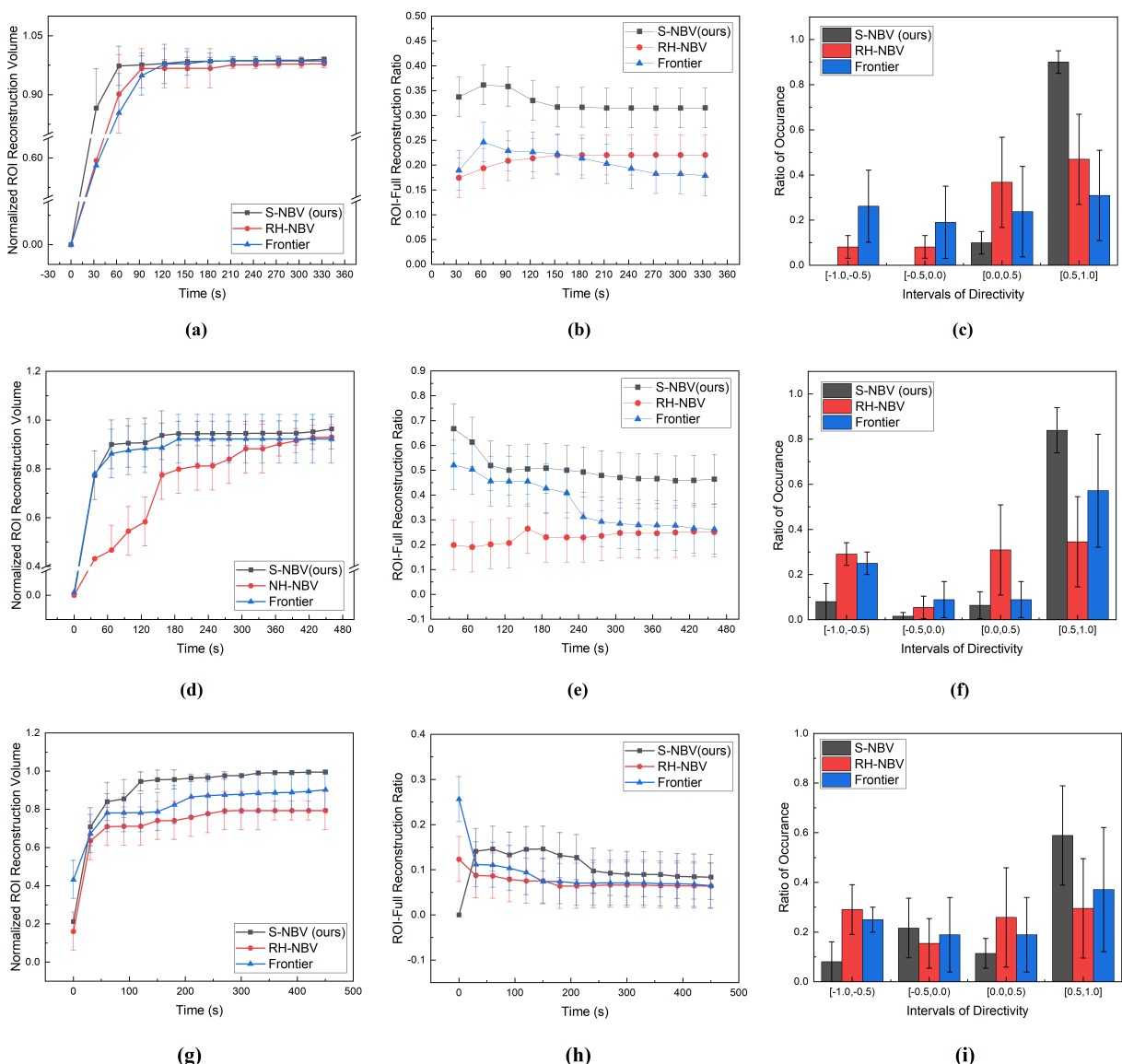

**Figure 3: Sub-figures (a), (b) are the normalized ROI reconstruction volume and ROI-to-full reconstruction volume ratio verse the simulation time in the Collapsed Room scene, sub-figure (c) represents the distribution of directivity during the completed experiment in the Collapsed Room scene. Sub-figures (d), (e), and (f) are the corresponding results in the Kitchen and Dining Room experiment. Sub-figures (g), (h), and(i) are the corresponding results in the Kitchen and Dining Room with Multiple Specified Objects. Each sub-figure presents the performance comparison between the proposed approach (S-NBV), RH-NBV [4] and the frontier-based approach [40].**

## 4.3 Experimental Results

The experiments in this study are conducted in two simulation scenes with three different settings in total. Compared to the smaller scene Collapsed Room, it typically takes longer for the UAV to locate the target in the larger one. In Figure 3(a) and (d), the target is well located within the first 30 s in each scenario. In complex scenes with more intricate structures, the proposed approach demonstrates

significant advantages over the RH-NBV approach [4], around 40% to 7% ahead at 60 s. Meanwhile, the frontier-based approach [40] also performed well in Figure 3(d) since it has a strong pattern of exploring along the large and continuous entity, such as walls. It also can be seen in the trajectory in Figure 4(h) that the target person in Kitchen and Dining Scene is located closer to the corner of the wall. However, the proposed approach still exhibits 12% to

**Figure 4: Original Scenes in Gazebo (the red square denotes the specified target): (a) Collapsed Room; (e) Kitchen and Dining Room; Sub-figures (b) and (f) show the motion trajectories planned by the proposed approach; (c) and (g) are the trajectories planned by RH-NBV [4]; (d) and (h) show the trajectories planned by the frontier-based approach [40]; The trajectories of different approaches are shown in the same global map, the trajectories of the proposed approach demonstrate the best target perceiving coverage around the target.**

**Table 2: Average Perspective Directivity of Entire Manoeuvre**

| Scene Name | S-NBV | RH-NBV | Frontier |
|---|---|---|---|
| Collapsed Room | 0.90516±0.04 | 0.44037±0.20 | 0.02282±0.34 |
| Kit & Din Room | 0.76042±0.02 | 0.13461±0.15 | 0.45207±0.40 |
| K&D Multi-Obj | 0.64037±0.2 | 0.10375±0.14 | 0.12442±0.26 |

3% advantages over the frontier-based approach at 60 s. And in both single target reconstruction progress, the ROI reconstruction volume increases every 30 s with the semantic-aware approach, i.e. we are progressively perceiving the ROI, while other approaches show less progress or even remain the same when the normalized reconstruction volume is greater than 0.9. The proposed approach finally achieves the ROI reconstruction progresses at 99.03% and 96.31%, while the other two planners stop at 97.86%, 98.58% in Collapsed Room and 93.12%, 92.27% in Kitchen and Dining Room, respectively. In the experiment involving multiple specified objects, the proposed method demonstrates a significant improvement in reconstruction progress, outperforming the existing planner by up to 23.45% within the first 120 seconds. Ultimately, it achieves a more detailed ROI reconstruction, surpassing the other methods by up to 20.19%, as illustrated in Figure 3(g).

In Figure 3(b) and (e), the proposed planner prioritizes the ROI reconstruction once the target is located, while other planners focus on perceiving other unknowns, which may belong to semantically redundant areas. The proposed approach achieves an average ROI-to-full ratio of 0.3268 and 0.5046 for each scene, respectively. In comparison, the RH-NBV and frontier-based approaches achieve averages of 0.2174, 0.2333 and 0.2228, 0.3586, respectively. For the multi-object scenario in Figure 3(h), our method shows less advantage in ROI-to-full ratio compared to the single object since more searching and target switching require more observations of the environment.

The distribution of view directivity during the entire manoeuvre is shown in Figure 3(c) and (f). The proposed planner exhibits stronger directionality and purposiveness towards the target, with a significant amount of perspective directivity (90% and 83.871%) falling in the interval [0.5, 1.0]. The proposed approach planned more views which have directivity in the range of [-1.0, 0.5) in the Kitchen and Dining scene because it takes more views to search for the target. In Figure 3(i), The multi-object directivity distribution spends more views switching between different targets, but the ratio of occurrence in [0.5,1.0] still dominates. In Table 14, the proposed approach shows a significant advantage over the other two planners by up to 0.88234 and 0.62581 in the average perspective directivity.

Figure 4 shows the two original scenes in Gazebo and planned trajectories by each planner, where the green, red and blue trajectories denote the planned ones by our method, the RH-NBV and the Frontier-based one in order. The green trajectories exhibit the maximum coverage of the viewing angles of the target, circling around the target within the reachable region. Compared to the trajectories of the other two in Figure 4(c), (d), (g), and (h), the proposed method also shows strong directionality and purposiveness towards the target and the target's surroundings evidently with its trajectory in Figure 4(b) and (f), while the others seem like "wandering aimlessly and enjoying freedom". Due to space limitations, more visualization results are presented in the supplementary material.

## 5 DISCUSSION AND FUTURE WORK

The proposed semantic-aware NBV scheme in this study demonstrates its advantages in search-and-acquisition manoeuvre under the complex environment over the existing informative path planners in the ROI reconstruction progress, ROI-to-full reconstruction volume ratio and perspective directivity. From the experimental results in Section 4, the RH-NBV planner also demonstrates good ROI reconstruction performance in the smaller scene but poor performance in the larger scene. The frontier-based planner exhibits a good ROI-to-full ratio at the beginning of the experiment, which profits from its pattern of pursuing the frontier voxels. After that, the planner intends to find the other unknowns, thus the ROI-to-full ratio drops. The significant difference here is that we are keen on perceiving the region we are interested in instead of pursuing unknown areas. More than 80% of the perspective directivity of the proposed approach falls in the interval [0.5, 1.0] in both scenes, while the other two distribute more average within four intervals. It means we are consistently looking towards the target's location once the target is well-located, while the other two are looking in all directions more evenly. The results also demonstrate the generalization potential of the proposed method by introducing the termination criterion to handle the multi-object search-and-acquisition. As the complexity of the manoeuvre increases, particularly when dealing with multiple objects, the proposed method offers a more exhaustive capture of the objects of interest.

However, there are some limitations to this work. First of all, both the planning and reconstruction processes are based on the same volumetric map. The choice of voxel size is a trade-off between reconstruction precision and planning efficiency, i.e. real-time smooth planning (e.g. around 4 to 7 seconds per planning) results in a compromise in the reconstruction precision. The second one is that the proposed approach is less aggressive in exploring or searching in the larger area than the frontier-based planner. Thus, it takes longer to locate the target as the area increases.

## 6 CONCLUSION

In this study, we presented a semantic-aware Next-Best-View aided by the multi-DoFs mobile system for autonomous visual perception under the complex and unknown environment. We formulate the novel semantic gain, combined with the conventional visibility gain in a unified form, to evaluate the "Next Best" view among the candidate views with the contribution of semantics. An adaptive strategy is introduced to control the mode switching between 'search' and 'acquisition' on the specific target under the challenging environment, and a termination criterion is designed to handle the target switching in multi-target visual acquisition. The capability of the proposed approach is demonstrated in three different settings in the simulation, achieving improvements of up to 27.13% for the ROI-to-full reconstruction volume ratio and a 0.88234 average perspective directivity. The planned motion trajectory is compared with the ones produced by existing planners, and a better target perceiving coverage is demonstrated evidently.

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
