# OpenReview forum: "Semantic-aware Next-Best-View for Multi-DoFs Mobile System in Search-and-Acquisition based Visual Perception"
_acmmm.org/ACMMM/2024/Conference — MM2024 Poster_

### Official Review · Reviewer_9Qvm · 2024-05-08

**Rating:** 4
**Confidence:** 2

**Summary:**

The authors propose a novel approach for efficient visual perception in unknown environments, integrating visibility gain and semantic gain to select the semantic-aware Next-Best-View. They design an adaptive strategy for a two-stage search-and-acquisition maneuver on multiple objects of interest using a Multi-DoFs mobile system. Evaluation metrics demonstrate significant improvements over existing methods, with simulation experiments confirming better target coverage in planned motion trajectories.

**Strengths:**

1. The idea of the method is clearly presented.
2. The motivation of the method is valid and the method seems to be effective and stable.

**Limitations:**

1. The method is only compared with traditional frontier-based strategy and RH-NBV, which is a method proposed in 2018, without comparison with most advanced researches. However, as a type of efficient and universal traditional scheme, this is acceptable. But whatever, the evaluation is not sufficient.
2. As an universal algorithm, more comparison on complexity analysis with traditional frontier-based strategy and RH-NBV should be included.

**Suitability:**

2

---

### Official Review · Reviewer_Jkvu · 2024-05-15

**Rating:** 2
**Confidence:** 2

**Summary:**

In this paper, authors formulate a novel information gain that integrates both visibility gain and semantic gain in a unified form to select the semantic-aware Next-Best-View. design an adaptive strategy with termination criterion to support a two-stage search-and-acquisition manoeuvre on multiple objects of interest aided by a Multi-DoFs mobile system. Additionally, authors introduce several semantically relevant reconstruction metrics to evaluate the performance of the proposed approach. Simulation experiments demonstrate the advantages of the proposed approach over existing methods

**Strengths:**

The authors' idea of using semantic information to select the Next-Best-View is excellent. They have designed an initial method and conducted experimental validation based on this concept, achieving correct and relatively promising results.

**Limitations:**

（1）The specific representation of semantics in the methodology is unclear. The authors need to describe, and possibly visualize, how semantic representation is connected to semantic gain.（2）In the experimental section, both quantitative and qualitative aspects fail to adequately explain the proposed method and comprehensively demonstrate its advantages.（3）The temporal performance, its impact on reconstruction results, and the execution of tasks are not evaluated, making it difficult to assess the specific improvements brought by the method.（4）The authors have not compared their method with others nor conducted the necessary ablation studies to demonstrate the overall and modular performance superiority of their method.（5）Simulated experiments alone are insufficient to comprehensively and objectively prove the effectiveness of the method

**Suitability:**

2

---

### Official Review · Reviewer_euw9 · 2024-05-18

**Rating:** 5
**Confidence:** 2

**Summary:**

This paper discusses the  efficient visual perception using mobile systems in unknown environments, particularly in operations such as search and rescue where swift and comprehensive perception of objects of interest is crucial. The authors point out that in complex environments, selecting the "Next-Best-View" (NBV) based solely on maximizing visibility gain is insufficient as it does not consider semantic information. Therefore, the paper proposes a novel information gain formulation that integrates both visibility gain and semantic gain to select the semantic-aware next best view. Additionally, an adaptive strategy with termination criteria is designed to support two-stage search-and-acquisition maneuvers for multiple objects of interest in complex environments. This framework includes two maps: an occupancy TSDF map and a labeled TSDF map, which provide information about the occupancy status of the environment and a detailed representation of the environment's geometry with semantics, respectively. The paper details the formulations for visibility gain and semantic gain and proposes an adaptive strategy and termination criteria to balance the search and acquisition phases.

**Strengths:**

1. This paper proposes a novel information gain formulation that integrates visibility gain and semantic gain.
2. A two-stage search and acquisition framework with adaptive strategies and termination criteria is designed for multiple objects of interest in complex environments.
3. The paper has a logical structure, gradually unfolding from problem description to related work, method introduction, experimental results, and discussion.
4. Clear visualization results, such as motion trajectories and scene layouts, are provided. Experiments conducted in the Gazebo simulator provide a simulation of a real environment, enhancing the credibility of the research findings.

**Limitations:**

1. The authors need to analyze the algorithmic complexity and conduct experimental comparative analysis on the time overhead. On resource-constrained mobile platforms, the resource and time overhead of the algorithm are of great importance.
2. The authors need to declare how semantic labels are obtained. The calculation of semantic gain depends on accurate semantic labeling, which may require complex image processing and machine learning models.
3. Although simulation experiments can provide well-controlled test conditions, the simulated environment may not fully replicate the complexity of the real world.
4. The reviewer know the challenges associated with rapidly establishing a real-world experimental platform. However, I would appreciate it if the authors could provide some guidance and analysis for constructing a real-world experiment.
5. The paper primarily focuses on static environments, and additional considerations may be needed for target tracking and view planning in dynamic settings. How should dynamic moving objects in the environment be handled?

**Suitability:**

3

---

### Official Review · Reviewer_x2H2 · 2024-05-24

**Rating:** 3
**Confidence:** 2

**Summary:**

This paper introduces a semantic-aware Next-Best-View (NBV) approach for visual perception in complex environments using multi-degree-of-freedom (Multi-DoFs) mobile systems. By integrating visibility gain and semantic gain into a unified information metric, the method optimizes viewpoint selection to enhance object detection and reconstruction. An adaptive strategy balances search and acquisition tasks. Simulation results show significant improvements in reconstruction accuracy and efficiency, making this approach highly effective for applications such as search and rescue operations.

**Strengths:**

1 The paper introduces a novel information gain formulation that integrates both visibility gain and semantic gain, which is a significant advancement over traditional methods that focus solely on visibility.
2 The paper compares the proposed method with RH-NBV and frontier-based approaches, demonstrating its superiority in various reconstruction metrics.
3 The formulation of the information gain metric combines semantic and visibility gains, enhancing the theoretical foundation for NBV strategies.

**Limitations:**

1 The integration of semantic gain and the adaptive strategy, while innovative, may introduce significant computational overhead. The paper does not provide detailed analysis or benchmarks on computational performance, which is crucial for real-time applications.
2 The method’s performance in larger and more dynamic environments is not thoroughly explored. The scalability to extensive real-world scenarios with more objects of interest or higher complexity remains uncertain.

**Suitability:**

1

---

### Meta-Review · Area_Chair_9xdi · 2024-07-02

**Recommendation:** Accept (Poster)
**Confidence:** 3

**Metareview:**

The paper presents a novel approach for efficient visual perception in unknown environments, integrating visibility gain and semantic gain to select the semantic-aware view.  The reviewers find the paper to be well-written; the experiments and their results are clearly presented.

On the other hand, the reviewers have risen the lack of some relevant information such as computational detail of the real-time application, algorithm complexity, ablation study. Even if most of these concerns seem to have been addressed in the rebuttal, there are still concerns among the reviewers who stated the need for improvements in the paper such as adding real-world experiments and dynamic scenarios but also explore deeper the correlation between segmentation accuracy and semantic gain and the need of more detailed and informative visualisation results.   If these concerns can be addressed in the camera-ready version, it would strengthen the paper and its contributions to the research community.  We encourage the authors to consider the comments of the reviewers and make an effort to improve the paper.